

# Clinical efficacy and safety of organ-sparing cystectomy: a systematic review and meta-analysis

Yi Zhang[1,2,*], Lei Peng[3,4,*], Yang Zhang[1], Hangxu Li[2], Songbei Li[5], Shaohua Zhang[4] and Jianguo Shi[1]

[1] Department of Urology, The First Affiliated Hospital of Jinzhou Medical University, Jinzhou Medical University, Jinzhou, Liaoning, China
[2] Department of Urology, The Third Affiliated Hospital of Jinzhou Medical University, Jinzhou Medical University, Jinzhou, Liaoning, China
[3] Institute of Urology, The Third Affiliated Hospital of Shenzhen University (Luohu Hospital Group), Shenzhen University, Shenzhen, Guangdong, China
[4] South China Hospital, Health Science Center, Shenzhen University, Shenzhen, Guangdong, China
[5] Department of Critical Care Medicine, Sichuan Provincial People's Hospital, University of Electronic Science and Technology of China, Chengdu, Sichuan, China
[*] These authors contributed equally to this work.

Corresponding authors
Shaohua Zhang, 22zsh@163.com
Jianguo Shi, sjg_cool@163.com

## ABSTRACT

**Background**. The clinical safety and efficacy of organ-sparing cystectomy (OSC) are subjects of ongoing debate, particularly concerning the potential increased risk of recurrence when retaining additional organs and its effectiveness in preserving sexual and urinary functions.

**Methods**. Adhering to the PRISMA 2020 statement and AMSTAR Guidelines, we conducted a systematic literature search up to February 2024 using PubMed, Embase, and Web of Science. The comparison focused on the clinical safety and effectiveness of OSC and standard radical cystectomy (SRC) in the treatment of bladder tumors. Our assessment covered several dimensions: Surgical safety outcomes (operation time, length of stay (LOS), estimated blood loss (EBL), and complications), oncological safety outcomes (recurrence rate, positive surgical margin rate, overall survival, and cancer-specific survival), and functional efficacy outcomes (daytime and nighttime urinary incontinence at 6 and 12 months, clean intermittent catheterization (CIC) rate, and erectile function within and after 1 year).

**Results**. The analysis included 19 eligible studies, encompassing 2,057 patients (1,189 OSC patients and 768 SRC patients). OSC demonstrated significant benefits in terms of erectile function and urinary continence without impacting CIC rates. No significant differences were observed in recurrence rate, positive surgical margin rate, overall survival, and cancer-specific survival. Furthermore, OSC and SRC were comparable in surgical safety outcomes, including operating time, LOS, EBL, and complications.

**Conclusions**. OSC offers notable advantages in erectile function and urinary continence. Despite limited clinical practice and potential selection bias, urologists may still consider OSC more based on their experience and specific patient factors.

## INTRODUCTION

Bladder cancer (BC), the tenth most common cancer globally, had an estimated 83,190 new cases in the United States in 2024, ranking fourth among new cancer cases in men and resulting in approximately 16,840 deaths (*Siegel, Giaquinto & Jemal, 2024*; *Siegel et al., 2023*). Radical cystectomy, the traditional gold standard for treating muscle-invasive bladder cancer or high-risk non-muscle-invasive bladder cancer, is linked with high complication and perioperative mortality rates (*Powles et al., 2022*; *Zheng et al., 2022*). This procedure is also strongly associated with postoperative erectile dysfunction, significantly affecting patients' quality of life (*Hautmann, De Petriconi & Volkmer, 2010*; *Zippe et al., 2004*). The standard radical cystectomy involves removing the bladder and surrounding adipose tissue, the distal ureters, and conducting a pelvic lymph node dissection. In male patients, it includes the prostate and seminal vesicles, while in female patients, it involves the uterus, part of the anterior vaginal wall, and the uterine adnexa (*Leow et al., 2019*).

Spitz and colleagues introduced the concept of organ-sparing cystectomy (OSC) as a modification of radical cystectomy with orthotopic neobladder reconstruction, targeting bladder non-urothelial tumors in young, sexually active men to preserve fertility and erectile function (*Spitz et al., 1999*). OSC, developed to mitigate the impact on quality of life, has evolved technologically. In males, OSC encompasses prostate-sparing cystectomy (preserving the prostate, seminal vesicles, vas deferens, and neurovascular bundles), capsule-sparing cystectomy (removing the bladder and prostate gland intact), seminal vesicles-sparing cystectomy (preserving the seminal vesicles, vas deferens, and neurovascular bundles), and nerve-sparing cystectomy. In females, OSC techniques are less described but include uterus-sparing cystectomy (preserving the uterus, fallopian tubes, ovaries, and anterior vaginal wall), vaginal-sparing cystectomy, and nerve-sparing cystectomy.

OSC aims to address potential quality of life improvements; however, the clinical safety and efficacy of OSC have been subjects of ongoing debate, particularly concerning the potential increased risk of recurrence when retaining additional organs and its effectiveness in preserving sexual and urinary functions. This article conducts a comprehensive and impartial meta-analysis of high-quality clinical literature on OSC, addressing gaps in the understanding of its clinical efficacy and safety.

## MATERIALS AND METHODS

### Protocol

This evidence-based analysis adheres to the PRISMA 2020 statement and AMSTAR guidelines (*Page et al., 2021*; *Shea et al., 2017*), ensuring a rigorous methodological approach. Our systematic review is registered on PROSPERO (CRD42023469647), reflecting our commitment to transparency and reproducibility.

### Literature search

We conducted a comprehensive literature search in PubMed, Embase, and Web of Science, focusing on studies published from the inception of these databases up to

February 2024. These studies compared non-organ-sparing and organ-sparing cystectomy in the treatment of bladder tumors, with a focus on clinical efficacy and safety. Our search terms were comprehensive and included key terms such as "Urinary Bladder Neoplasms", "Cystectomy", "Prostate", "Capsule", "Seminal Vesicles", "Neurovascular Bundle", "NVB", "Nerve", "Uterus", "Fallopian Tubes", "Ovaries", "Vagina", "Sparing", "Protect", "Reserve", and "Preserve". Due to the involvement of organ-sparing techniques in both prostate cancer and uterine cancer, we excluded literature related to prostate cancer and uterine cancer. The complete search strategy is detailed in Table S1. Moreover, we manually reviewed references of all eligible studies and had two researchers (YZ and LP) independently evaluate the included studies, resolving any disagreements through consensus.

### Identification of eligible studies

Our inclusion criteria were stringent to ensure study relevance and quality:
(1) We included randomized control, cohort, or case-control studies.
(2) The studies had to involve men or women with bladder tumors, including various organ-sparing procedures specific to each gender.
(3) The comparison was between organ-sparing cystectomy (OSC) and standard radical cystectomy (SRC), focusing on preserving or not preserving pelvic organs.
(4) We assessed both clinical safety (operation time, hospital stay, EBL, complications) and oncological safety (surgical margins, recurrence rate, OS, CSS). Clinical efficacy was evaluated in terms of erectile function, urinary incontinence, and CIC rate over specific time frames.
(5) Only studies with sufficient data to compute odds ratios (OR) or weighted mean differences (WMD) were considered.

### Data Extraction

Data extraction was independently conducted by two researchers (YZ and LP), with a third researcher (JS) resolving any disagreements to make the final decision. We extracted the following data from the included studies: first author, publication year, study period, study design, sample size, age, clinical bladder stage, pathological bladder stage, type of surgery, urinary diversion, pathological N stage, follow-up duration, operative time, hospital stay, estimated blood loss (EBL), complications, recurrence rate, positive surgical margins rate, overall survival (OS), cancer-specific survival (CSS), erectile function within and after 1 year, and daytime and nighttime urinary incontinence at 6 and 12 months postoperatively, and CIC rate.

For evaluating urinary continence and potency, we applied standardized criteria in the absence of definitions from individual studies: urinary continence as needing ≤1 pad during day or night, and potency defined by either an adequate erection for intercourse or an International Index of Erectile Function (IIEF) score ≥20. This meta-analysis did not differentiate between types of ORC and imposed no language restrictions. For continuous variables reported as median and range, we calculated mean ± standard deviation using established methods (Luo et al., 2018; Wan et al., 2014). We contacted authors for missing data when necessary.

## Quality assessment

Quality assessment varied by study design. Randomized Controlled Trials (RCTs) were evaluated using the Cochrane risk of bias 2.0 tool (*Sterne et al., 2019*), while cohort and case-control studies were assessed *via* the Newcastle-Ottawa Scale (NOS) (*Wells et al.*). Studies scoring 7–9 on the NOS were deemed high quality (*Gan et al., 2023*). Two researchers (YZ and LP) independently evaluated the evidence quality and resolved differences through discussion.

## Statistical analysis

We utilized Review Manager 5.4 and STATA 17.0 for statistical analysis (*Yong & Guang, 2016*), employing Engauge Digitizer 4.1 for image data extraction. Binary variables were analyzed using OR with 95% confidence intervals (CI), and continuous data were assessed using weighted mean differences (WMD) and 95% CI (*Wan et al., 2014*). Heterogeneity was evaluated using Cochrane Q test and I2 statistics (*Higgins & Thompson, 2002*), adopting a random-effects model for significant heterogeneity ($p < 0.05$ or $I^2 > 50\%$). Statistical significance was set at $p < 0.05$. Publication bias was assessed using Egger's test and funnel plots (*Egger et al., 1997*). The GRADE system provided a structured framework for evaluating the quality of study outcomes, allowing for a thorough assessment of the strength and limitations of the evidence. This systematic grading process aids clinicians and decision-makers in developing more appropriate treatment plans and policies based on the varying quality of the evidence (*Guyatt et al., 2008*).

## Subgroup analyses and sensitivity analysis

Subgroup analyses were conducted based on factors like surgery type in OSC, study design, and assessment modality for continence and erectile function. Sensitivity analyses evaluated the impact of individual studies on outcomes with significant heterogeneity ($I^2 > 50\%$).

# RESULTS

## Literature search and study characteristics

Our systematic search, detailed in Fig. 1, yielded a comprehensive collection of 1,280 articles from PubMed, Embase, Web of Science, and citation searches. After removing duplicates, we screened 972 titles and abstracts, ultimately selecting 19 full-text articles for pooled analysis, involving 2,057 patients (1,189 ORC *vs* 768 SRC) (*Abdelaziz et al., 2019*; *Bai et al., 2019*; *Basiri et al., 2012*; *Chen & Chiang, 2017*; *Cheng et al., 2022*; *De Vries et al., 2009*; *El-Bahnasawy, Gomha & Shaaban, 2006*; *Furrer et al., 2018*; *Hekal et al., 2009*; *Huang et al., 2019*; *Kessler et al., 2004*; *Kwon et al., 2018*; *Moon, Park & Ahn, 2005*; *Park et al., 2022*; *Patel et al., 2022*; *Turner et al., 1997*; *Vilaseca et al., 2013*; *Vogt et al., 2021*; *Wang, Luo & Chen, 2008*). These studies comprised five prospective cohort studies (*De Vries et al., 2009*; *Furrer et al., 2018*; *Hekal et al., 2009*; *Kessler et al., 2004*; *Turner et al., 1997*), 13 retrospective cohort studies (*Abdelaziz et al., 2019*; *Bai et al., 2019*; *Basiri et al., 2012*; *Chen & Chiang, 2017*; *Cheng et al., 2022*; *El-Bahnasawy, Gomha & Shaaban, 2006*; *Huang et al., 2019*; *Kessler et al., 2004*; *Kwon et al., 2018*; *Moon, Park & Ahn, 2005*; *Park et al., 2022*; *Patel et al., 2022*; *Vilaseca et al., 2013*; *Vogt et al., 2021*; *Wang, Luo & Chen, 2008*), and one

prospective randomized study (*Abdelaziz et al., 2019*). The characteristics and quality scores of the included studies (median score 8, range 6–9) are summarized in Table 1, with 17 studies classified as high quality (*Abdelaziz et al., 2019*; *Bai et al., 2019*; *Chen & Chiang, 2017*; *Cheng et al., 2022*; *De Vries et al., 2009*; *Furrer et al., 2018*; *Hekal et al., 2009*; *Huang et al., 2019*; *Kessler et al., 2004*; *Kwon et al., 2018*; *Moon, Park & Ahn, 2005*; *Park et al., 2022*; *Patel et al., 2022*; *Turner et al., 1997*; *Vilaseca et al., 2013*; *Vogt et al., 2021*; *Wang, Luo & Chen, 2008*). Comprehensive quality assessments of all studies are available in Tables S2, and S3 delineates the clinical and pathological characteristics of the studies included. Comprehensive analysis indicates that OSC offers significant advantages over SRC in improving postoperative erectile function and urinary continence, while maintaining comparable surgical and oncological safety between the two groups.

## Surgical safety
### Operating time
Analysis of operating time from eight studies involving 556 patients (271 OSC *vs* 285 SRC) revealed no significant differences between groups (WMD: −16.99 ; 95% CI:-37.91, 3.93; $p = 0.11$) (*Abdelaziz et al., 2019*; *Bai et al., 2019*; *Cheng et al., 2022*; *Huang et al., 2019*; *Kwon et al., 2018*; *Moon, Park & Ahn, 2005*; *Vogt et al., 2021*; *Wang, Luo & Chen, 2008*). However, there was notable heterogeneity ($I^2 = 80\%$, $p < 0.0001$) (Fig. 2A). The funnel plot (Fig. S2A) and Egger's test ($p = 0.845$) indicated no apparent bias.

### Length of stay
Data from five studies on length of stay, covering 308 patients (151 OSC *vs* 157 SRC) (*Bai et al., 2019*; *Cheng et al., 2022*; *Huang et al., 2019*; *Kwon et al., 2018*; *Moon, Park & Ahn, 2005*), showed no significant differences (WMD: 0.93; 95% CI: −0.54, 2.39; $p = 0.21$) with moderate heterogeneity ($I^2 = 43\%$, $p = 0.13$) (Fig. 2B).

### Estimated blood loss
Estimated blood loss was assessed in six studies with 473 patients (276 OSC *vs* 197 SRC), showing no significant differences between groups (WMD: −63.73; 95% CI: −142.70, 15.25; $p = 0.11$) (*Bai et al., 2019*; *Cheng et al., 2022*; *Huang et al., 2019*; *Kwon et al., 2018*; *Patel et al., 2022*; *Wang, Luo & Chen, 2008*), despite high heterogeneity ($I^2 = 88\%$, $p < 0.00001$) (Fig. 2C).

### Complications
Five studies, involving 389 patients (182 OSC *vs* 207 SRC), reported on complications (*Bai et al., 2019*; *Cheng et al., 2022*; *Huang et al., 2019*; *Kwon et al., 2018*; *Vogt et al., 2021*), revealing no significant differences between OSC and SRC (OR: 1.06; 95% CI: 0.50, 2.24; $p = 0.88$), but with notable heterogeneity ($I^2 = 63\%$, $p = 0.03$) (Fig. 2D).

## Oncological safety
### Recurrence rate
Recurrence rates were analyzed in nine studies with 847 patients (440 OSC *vs* 407 SRC), showing no significant differences (OR: 0.80; 95% CI: 0.56, 1.15; $p = 0.23$) (*Abdelaziz et al.,*

Zhang et al. (2024), *PeerJ*, DOI 10.7717/peerj.18427

**Table 1** Baseline characteristics of include studies and methodological assessment.

| Authors | Country | Patients (ORC) | Patients (SRC) | Study period | Follow-up duration (median/months) | | Study design | Type of surgery (RC) | NOS |
|---|---|---|---|---|---|---|---|---|---|
| *Park et al. (2022)* | Korea | 40 | 46 | 2009-2020 | 25.9(Iqr4.7-85.3) | 18.8(Iqr2.6-131.9) | Retro | USC vs SRC | 8 |
| *Chen & Chiang (2017)* | China | 14 | 11 | 2007-2015 | 51.14 | 73.82 | Retro | PSC vs SRC | 8 |
| *Abdelaziz et al. (2019)* | Egypt | 45 | 51 | 2014-2016 | 24 | | RCT | CSC vs SRC | (RCT) Low risk |
| *Vilaseca et al. (2013)* | Spain | 11 | 33 | 2006-2009 | 21 | | Retro | NSC vs SRC | 7 |
| *Kwon et al. (2018)* | Korea | 15 | 23 | 2009-2014 | 80 | 43 | Retro | NSC vs SRC | 7 |
| *Moon, Park & Ahn (2005)* | Korea | 17 | 18 | 1999-2003 | 16.1(range 6–27) | 17.9(range 6–44) | Retro | NSC vs SRC | 8 |
| *Furrer et al. (2018)* | Switzerland | 156 | 24 | 1985-2007 | Uni 174(Iqr152-209) Bi 163 (Iqr132-203) | 177 (Iqr161-232) | Pros | NSC vs SRC | 8 |
| *Cheng et al. (2022)* | China | 11 | 22 | 2018–2019 | 17(range 12–22) | | Retro | NSC vs SRC | 8 |
| *Basiri et al. (2012)* | Iran | 23 | 27 | none | 39 | 35 | Retro | PSC vs SRC | 6 |
| *Vogt et al. (2021)* | Germany | 48 | 68 | 2012–2019 | none | none | Retro | NSC vs SRC | 7 |
| *De Vries et al. (2009)* | Netherlands | 63 | 63 | 1994–2006 | 56 | 76 | Pros | PSC vs SRC | 8 |
| *Hekal et al. (2009)* | Egypt | 21 | 24 | 2003–2005 | 16.4(range 12–24) | | Pros | NSC vs SRC | 8 |
| *Patel et al. (2022)* | The USA | 188 | 101 | 2000-2020 | none | none | Retro | USC vs SRC | 9 |
| *Turner et al. (1997)* | Switzerland | 116 | 49 | 1985–1996 | Uni 30(range 4–101) Bi 31(range 6–86) | 46(range 3–134) | Pros | NSC vs SRC | 7 |
| *Bai et al. (2019)* | China | 45 | 45 | 2007–2017 | 34.0(Iqr8.5-54.0) | 38.0(Iqr15.0-49.0) | Retro | USC vs SRC | 9 |
| *El-Bahnasawy, Gomha & Shaaban (2006)* | Egypt | 30 | 30 | none | 38.8 ± 19.2[*] | 42.9 ± 26.9[*] | Retro | NSC vs SRC | 6 |
| *Kessler et al. (2004)* | Switzerland | 256 | 75 | 1985–2003 | 2.6(Iqr1-6) | | Pros | NSC vs SRC | 7 |
| *Wang, Luo & Chen (2008)* | China | 27 | 9 | 2000–2006 | 3–84 | | Retro | CSC vs SRC | 7 |

Zhang et al. (2024), *PeerJ*, DOI 10.7717/peerj.18427

**Table 1** (*continued*)

| Authors | Country | Patients (ORC) | Patients (SRC) | Study period | Follow-up duration (median/months) | Study design | Type of surgery (RC) | NOS |
|---|---|---|---|---|---|---|---|---|
| *Huang et al. (2019)* | China | 63 | 49 | 2006–2017 | 36(Iqr16-69) | Retro | USC vs SRC | 8 |

**Notes.**

*Means + standard deviation

Uni, Unilateral; Bi, Bilateral; Retro, Retrospective; Pros, Prospective; SRC, standard radical cystectomy; organ sparing cystectomy; USC, uterus sparing cystectomy; PSC, prostate sparing cystectomy; CSC, capsule sparing cystectomy; NSC, nerve sparing cystectomy; NOS, Newcastle–Ottawa Scale.

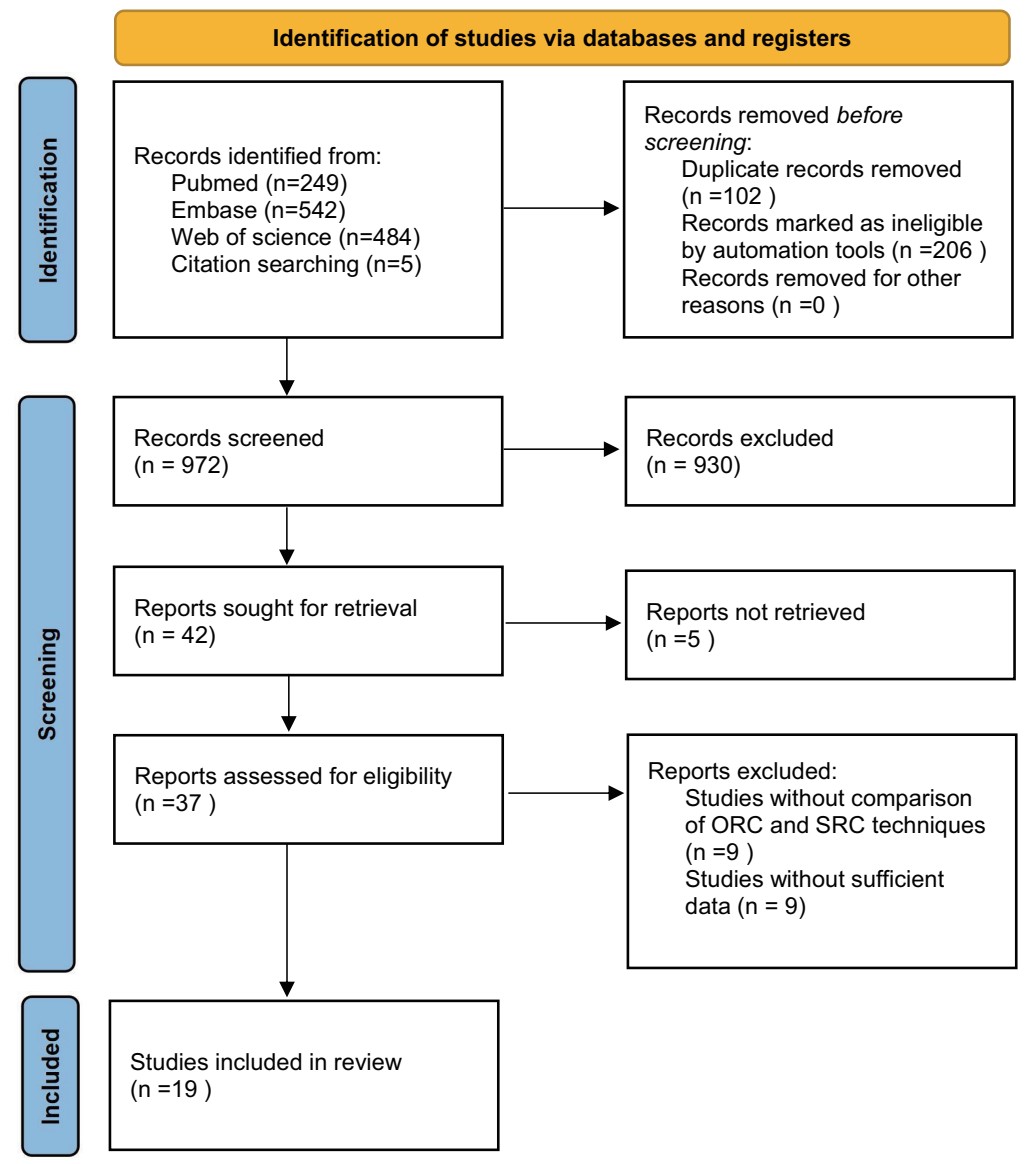

**Figure 1** Flowchart of the systematic search and selection process.

2019; *Bai et al., 2019*; *Basiri et al., 2012*; *Cheng et al., 2022*; *De Vries et al., 2009*; *Hekal et al., 2009*; *Park et al., 2022*; *Patel et al., 2022*; *Vilaseca et al., 2013*), with negligible heterogeneity ($I^2 = 0\%$, $p = 0.71$) (Fig. 3A). The funnel plot (Fig. S2B) and Egger's test ($p = 0.519$) indicated no publication bias.

### Positive surgical margin rate

Positive surgical margins were evaluated in six studies involving 762 patients (413 OSC *vs* 349 SRC), with no significant differences found (OR: 0.73; 95% CI: 0.45, 1.20; $p = 0.22$)

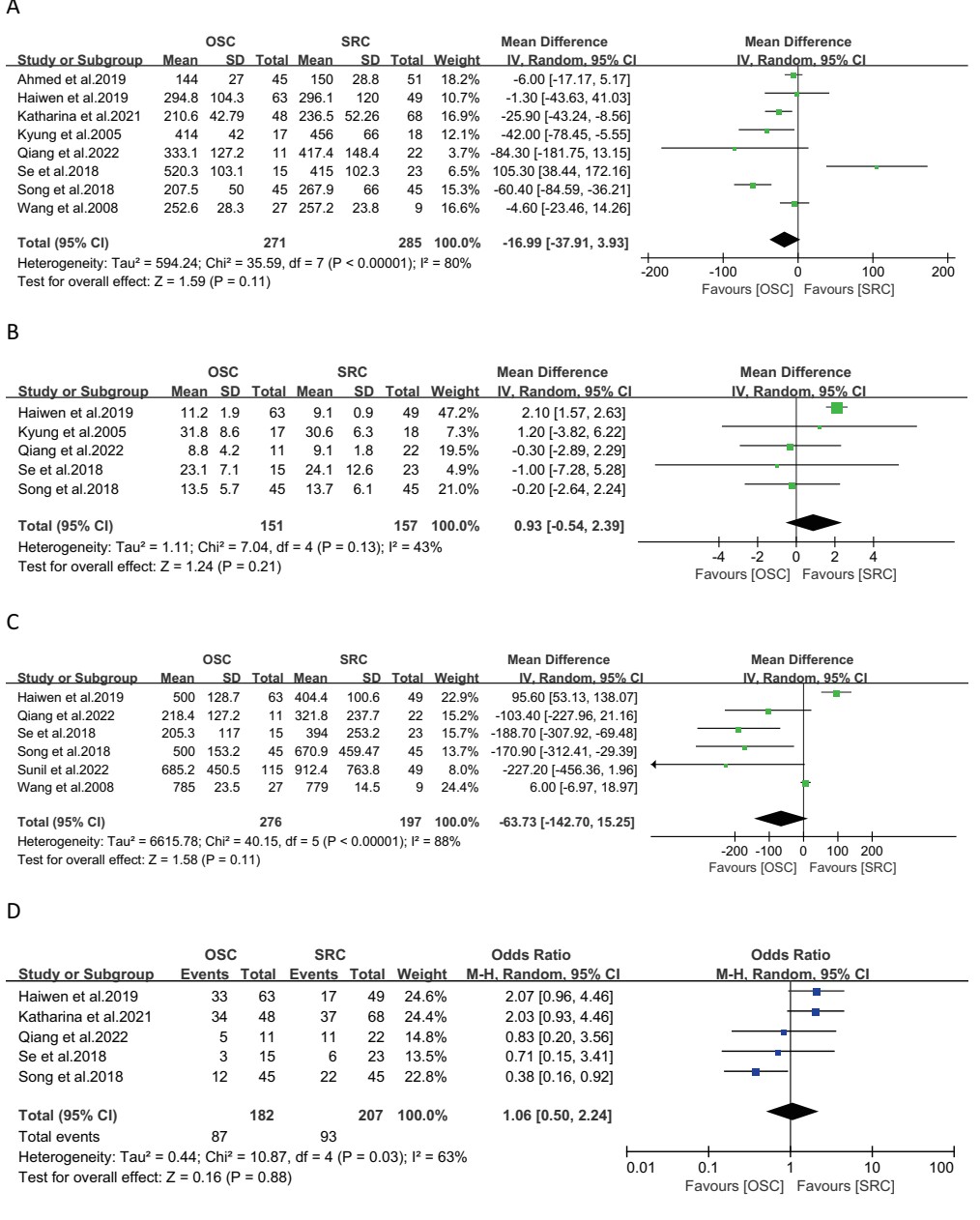

**Figure 2** Forest plots of surgical safety: (A) operating time, (B) length of stay, (C) estimated blood loss, (D) complications. (A) *Abdelaziz et al. (2019)*; *Huang et al. (2019)*; *Vogt et al. (2021)*; *Moon, Park & Ahn (2005)*; *Cheng et al. (2022)*; *Kwon et al. (2018)*; *Bai et al. (2019)*; *Wang, Luo & Chen (2008)*. (B) *Huang et al. (2019)*; *Moon, Park & Ahn (2005)*; *Cheng et al. (2022)*; *Kwon et al. (2018)*; *Bai et al. (2019)*. (C) *Huang et al. (2019)*; *Cheng et al. (2022)*; *Kwon et al. (2018)*; *Bai et al. (2019)*; *Patel et al. (2022)*; *Wang, Luo & Chen (2008)*. (D) *Huang et al. (2019)*; *Vogt et al. (2021)*; *Cheng et al. (2022)*; *Kwon et al. (2018)*; *Bai et al. (2019)*.

(*Cheng et al., 2022*; *De Vries et al., 2009*; *Huang et al., 2019*; *Park et al., 2022*; *Patel et al., 2022*; *Vogt et al., 2021*), with no significant heterogeneity ($I^2 = 0\%$, $p = 0.67$) (Fig. 3B).

A

| Study or Subgroup | OSC Events | OSC Total | SRC Events | SRC Total | Weight | Odds Ratio M-H, Fixed, 95% CI |
|---|---|---|---|---|---|---|
| Abbas et al.2012 | 11 | 18 | 11 | 20 | 6.0% | 1.29 [0.35, 4.69] |
| Ahmed et al.2019 | 1 | 48 | 5 | 56 | 6.7% | 0.22 [0.02, 1.93] |
| Ihab et al.2008 | 0 | 21 | 0 | 24 | | Not estimable |
| Jae et al.2022 | 12 | 40 | 15 | 46 | 14.4% | 0.89 [0.35, 2.21] |
| Qiang et al.2022 | 0 | 11 | 0 | 22 | | Not estimable |
| Remco et al.2009 | 5 | 63 | 10 | 63 | 13.5% | 0.46 [0.15, 1.42] |
| Song et al.2018 | 13 | 45 | 11 | 45 | 11.5% | 1.26 [0.49, 3.20] |
| Sunil et al.2022 | 48 | 183 | 31 | 100 | 43.5% | 0.79 [0.46, 1.35] |
| Vilaseca et al.2013 | 2 | 11 | 7 | 31 | 4.4% | 0.76 [0.13, 4.38] |
| | | | | | | |
| Total (95% CI) | | 440 | | 407 | 100.0% | 0.80 [0.56, 1.15] |
| Total events | 92 | | 90 | | | |

Heterogeneity: Chi² = 3.76, df = 6 (P = 0.71); I² = 0%
Test for overall effect: Z = 1.21 (P = 0.23)

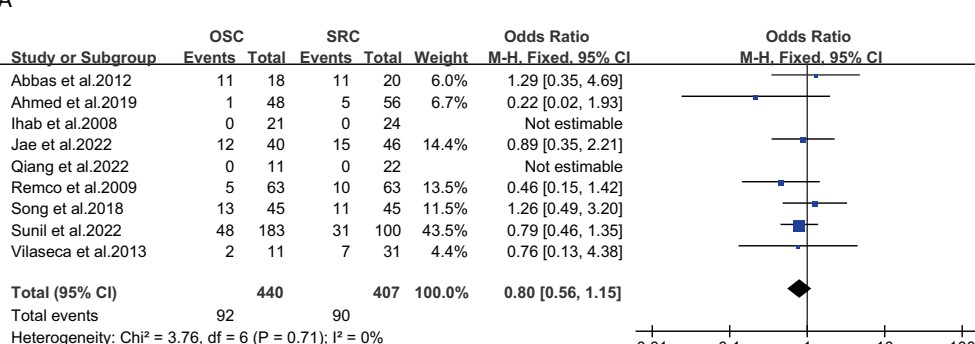

B

| Study or Subgroup | OSC Events | OSC Total | SRC Events | SRC Total | Weight | Odds Ratio M-H, Fixed, 95% CI |
|---|---|---|---|---|---|---|
| Haiwen et al.2019 | 3 | 63 | 2 | 49 | 5.9% | 1.18 [0.19, 7.32] |
| Jae et al.2022 | 6 | 40 | 4 | 46 | 8.7% | 1.85 [0.48, 7.10] |
| Katharina et al.2021 | 2 | 48 | 3 | 68 | 6.6% | 0.94 [0.15, 5.86] |
| Qiang et al.2022 | 0 | 11 | 1 | 22 | 2.7% | 0.62 [0.02, 16.56] |
| Remco et al.2009 | 2 | 63 | 5 | 63 | 13.4% | 0.38 [0.07, 2.04] |
| Sunil et al.2022 | 24 | 188 | 20 | 101 | 62.7% | 0.59 [0.31, 1.14] |
| | | | | | | |
| Total (95% CI) | | 413 | | 349 | 100.0% | 0.73 [0.45, 1.20] |
| Total events | 37 | | 35 | | | |

Heterogeneity: Chi² = 3.16, df = 5 (P = 0.67); I² = 0%
Test for overall effect: Z = 1.23 (P = 0.22)

C

| Study or Subgroup | OSC Events | OSC Total | SRC Events | SRC Total | Weight | Odds Ratio M-H, Fixed, 95% CI |
|---|---|---|---|---|---|---|
| Abbas et al.2012 | 11 | 23 | 8 | 27 | 6.9% | 2.18 [0.68, 6.96] |
| Haiwen et al.2019 | 39 | 63 | 29 | 49 | 22.3% | 1.12 [0.52, 2.41] |
| Jae et al.2022 | 28 | 40 | 33 | 46 | 16.5% | 0.92 [0.36, 2.34] |
| Se et al.2018 | 13 | 15 | 18 | 23 | 3.4% | 1.81 [0.30, 10.80] |
| Song et al.2018 | 19 | 45 | 22 | 45 | 22.8% | 0.76 [0.33, 1.75] |
| Sunil et al.2022 | 31 | 90 | 22 | 75 | 28.2% | 1.27 [0.65, 2.45] |
| | | | | | | |
| Total (95% CI) | | 276 | | 265 | 100.0% | 1.14 [0.80, 1.64] |
| Total events | 141 | | 132 | | | |

Heterogeneity: Chi² = 2.64, df = 5 (P = 0.76); I² = 0%
Test for overall effect: Z = 0.73 (P = 0.47)

D

| Study or Subgroup | OSC Events | OSC Total | SRC Events | SRC Total | Weight | Odds Ratio M-H, Fixed, 95% CI |
|---|---|---|---|---|---|---|
| Jae et al.2022 | 28 | 40 | 35 | 46 | 20.2% | 0.73 [0.28, 1.91] |
| Remco et al.2009 | 42 | 63 | 40 | 63 | 27.6% | 1.15 [0.55, 2.39] |
| Se et al.2018 | 13 | 15 | 20 | 23 | 4.4% | 0.97 [0.14, 6.65] |
| Song et al.2018 | 33 | 45 | 28 | 45 | 15.5% | 1.67 [0.68, 4.08] |
| Sunil et al.2022 | 36 | 72 | 31 | 71 | 32.3% | 1.29 [0.67, 2.49] |
| | | | | | | |
| Total (95% CI) | | 235 | | 248 | 100.0% | 1.18 [0.81, 1.74] |
| Total events | 152 | | 154 | | | |

Heterogeneity: Chi² = 1.64, df = 4 (P = 0.80); I² = 0%
Test for overall effect: Z = 0.86 (P = 0.39)

**Figure 3** **Forest plots of oncological safety: (A) recurrence rate, (B) positive surgical margin rate, (C) overall survival, (D) cancer specific survival.** (A) *Basiri et al. (2012)*; *Abdelaziz et al. (2019)*; *Hekal et al. (2009)*; *Park et al. (2022)*; *Cheng et al. (2022)*; *De Vries et al. (2009)*; *Bai et al. (2019)*; *Patel et al. (2022)*; *Vilaseca et al. (2013)*. (B) *Huang et al. (2019)*; *Park et al. (2022)*; *Vogt et al. (2021)*; *Cheng et al. (2022)*; *De Vries et al. (2009)*; *Patel et al. (2022)*. (C) *Basiri et al. (2012)*; *Huang et al. (2019)*; *Park et al. (2022)*; *Kwon et al. (2018)*; *Bai et al. (2019)*; *Patel et al. (2022)*. (D) *Park et al. (2022)*; *De Vries et al. (2009)*; *Kwon et al. (2018)*; *Bai et al. (2019)*; *Patel et al. (2022)*.

*Overall survival and cancer-specific survival*

Five-year survival rates, assessed in six articles with 273 patients (OSC 141, SRC 132), showed similar outcomes for both groups (OR: 1.14; 95% CI: 0.80, 1.64; $p = 0.47$) (*Bai et al., 2019*; *Basiri et al., 2012*; *Huang et al., 2019*; *Kwon et al., 2018*; *Park et al., 2022*; *Patel et al., 2022*), with no significant heterogeneity ($I^2 = 0\%$, $p = 0.76$) (Fig. 3C). Cancer-specific survival, analyzed in five studies involving 483 patients (235 OSC *vs* 248 SRC), also showed similar results (OR: 1.18; 95% CI: 0.81, 1.74; $p = 0.39$) (Fig. 3D) (*Bai et al., 2019*; *De Vries et al., 2009*; *Kwon et al., 2018*; *Park et al., 2022*; *Patel et al., 2022*), with negligible heterogeneity ($I^2 = 0\%$, $p = 0.80$).

## Outcome measures

*Daytime and nighttime urinary incontinence at 6 months*

Analysis from eight studies on daytime urinary incontinence at 6 months post-surgery (932 patients: 655 OSC *vs* 277 SRC) indicated a significantly increased risk of incontinence in the SRC group (OR: 4.19; 95% CI: 2.26, 7.79; $p < 0.00001$) (Fig. 4A) (*Abdelaziz et al., 2019*; *Cheng et al., 2022*; *Furrer et al., 2018*; *Kessler et al., 2004*; *Park et al., 2022*; *Turner et al., 1997*; *Vilaseca et al., 2013*; *Wang, Luo & Chen, 2008*), with moderate heterogeneity ($I^2 = 55\%$, $p = 0.03$). Nighttime continence also showed a similar increased risk in eight studies (933 patients: 656 OSC *vs* 277 SRC), with significant results (OR: 3.14; 95% CI: 1.55, 6.34; $p = 0.001$) (Fig. 4C) (*Abdelaziz et al., 2019*; *Cheng et al., 2022*; *Furrer et al., 2018*; *Kessler et al., 2004*; *Park et al., 2022*; *Turner et al., 1997*; *Vilaseca et al., 2013*; *Wang, Luo & Chen, 2008*), with high heterogeneity ($I^2 = 68\%$, $p = 0.003$). In both analyses, neither the funnel plot (Figs. S2C–S2D) nor Egger's test (daytime:p =0.176;nighttime:p = 0.191) suggested publication bias.

*Daytime and nighttime urinary incontinence at 12 months*

Seven studies on daytime urinary incontinence at 12 months post-surgery (890 patients: 628 OSC *vs* 262 SRC) revealed a significantly increased risk of incontinence in the SRC group (OR: 4.20; 95% CI: 2.68, 6.59; $p < 0.00001$) (Fig. 4B) (*Abdelaziz et al., 2019*; *Chen & Chiang, 2017*; *Cheng et al., 2022*; *El-Bahnasawy, Gomha & Shaaban, 2006*; *Furrer et al., 2018*; *Kessler et al., 2004*; *Turner et al., 1997*), with low heterogeneity ($I^2 = 10\%$, $p = 0.35$). Nighttime incontinence analysis showed a similar trend (OR: 2.65; 95% CI: 1.43, 4.93; $p = 0.002$) (Fig. 4D), albeit with high heterogeneity ($I^2 = 63\%$, $p = 0.01$).

*CIC rate*

In five studies involving 389 patients (265 OSC *vs* 124 SRC), CIC rates showed no significant differences (OR: 1.02; 95% CI: 0.16, 6.44; $p = 0.98$) (Fig. 4E) (*Abdelaziz et al., 2019*; *Basiri et al., 2012*; *Chen & Chiang, 2017*; *Furrer et al., 2018*; *Park et al., 2022*). However, there was significant heterogeneity ($I^2 = 83\%$, $p = 0.001$).

*Erectile function within and after 1 year*

Short-term (<1 year) erectile function improvement in the OSC group was significant, as reported in four studies involving 498 patients (345 OSC *vs* 153 SRC) (OR: 27.52; 95% CI: 2.58, 294.07; $p = 0.006$) (Fig. 4F) (*Abdelaziz et al., 2019*; *Kessler et al., 2004*; *Moon, Park*

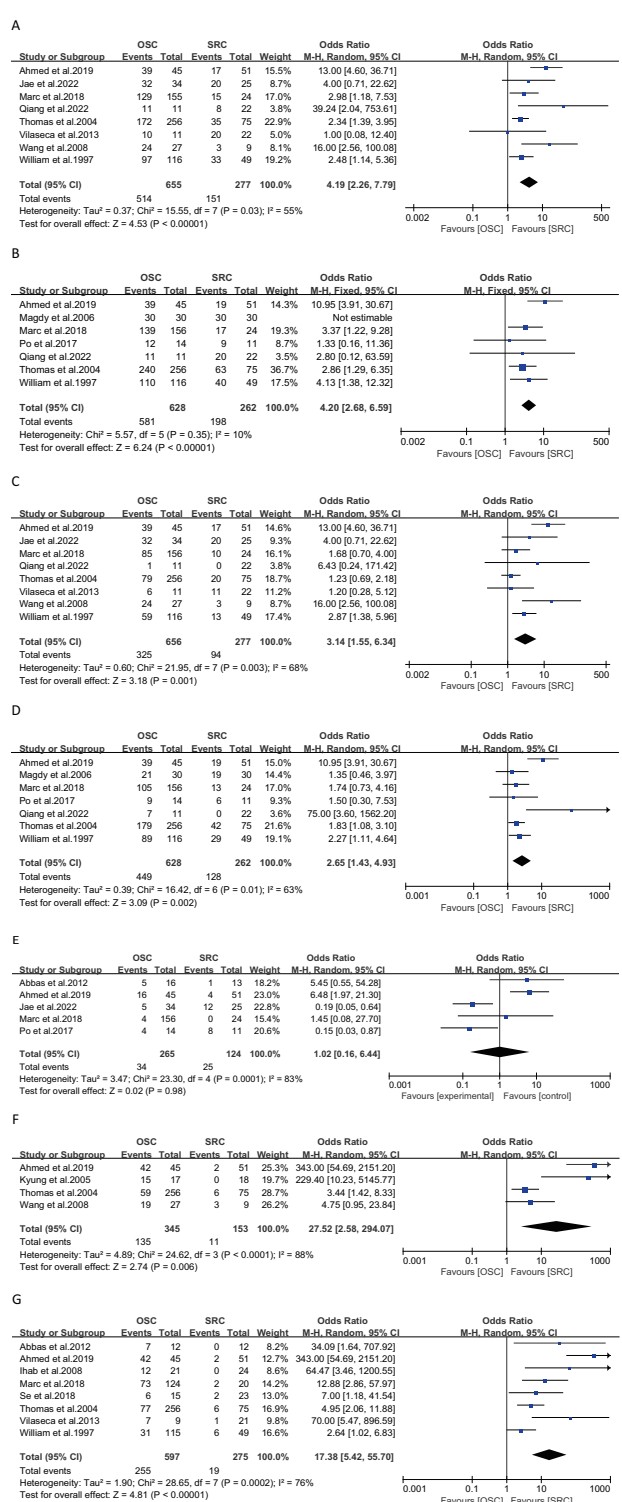

**Figure 4** Forest plots of clinical efficacy: (A) daytime urinary incontinence at 6 months, (B) daytime urinary incontinence at 12 months, (C) nighttime urinary incontinence at 6 months , (D) nighttime urinary incontinence at 12 months, (E) CIC rate, (F) erectile function within 1 year, (G) erectile function after 1 year. (continued on next page...)

**Figure 4 (…continued)**
(A) *Abdelaziz et al. (2019)*; *Park et al. (2022)*; *Furrer et al. (2018)*; *Cheng et al. (2022)*; *Kessler et al. (2004)*; *Vilaseca et al. (2013)*; *Wang, Luo & Chen (2008)*; *Turner et al. (1997)*.(B) *Abdelaziz et al. (2019)*; *El-Bahnasawy, Gomha & Shaaban (2006)*; *Furrer et al. (2018)*; *Chen & Chiang (2017)*; *Cheng et al. (2022)*; *Kessler et al. (2004)*; *Turner et al. (1997)*. (C) *Abdelaziz et al. (2019)*; *Park et al. (2022)*; *Furrer et al. (2018)*; *Cheng et al. (2022)*; *Kessler et al. (2004)*; *Vilaseca et al. (2013)*; *Wang, Luo & Chen (2008)*; *Turner et al. (1997)*. (D) *Abdelaziz et al. (2019)*; *El-Bahnasawy, Gomha & Shaaban (2006)*; *Furrer et al. (2018)*; *Chen & Chiang (2017)*; *Cheng et al. (2022)*; *Kessler et al. (2004)*; *Turner et al. (1997)*. (E) *Basiri et al. (2012)*; *Abdelaziz et al. (2019)*; *Park et al. (2022)*; *Furrer et al. (2018)*; *Chen & Chiang (2017)*. (F) *Abdelaziz et al. (2019)*; *Moon, Park & Ahn (2005)*; *Kessler et al. (2004)*; *Wang, Luo & Chen (2008)*. (G) *Basiri et al. (2012)*; *Abdelaziz et al. (2019)*; *Hekal et al. (2009)*; *Furrer et al. (2018)*; *Kwon et al. (2018)*; *Kessler et al. (2004)*; *Vilaseca et al. (2013)*; *Turner et al. (1997)*.

*& Ahn, 2005*; *Wang, Luo & Chen, 2008*), with high heterogeneity ($I^2 = 88\%$, $p < 0.0001$). Long-term ($\geq 1$ year) erectile function also showed significant improvement in the OSC group, as indicated in eight studies with 872 patients (597 OSC *vs* 275 SRC) (OR: 17.38; 95% CI: 5.42, 55.70; $p < 0.00001$) (*Abdelaziz et al., 2019*; *Basiri et al., 2012*; *Furrer et al., 2018*; *Hekal et al., 2009*; *Kessler et al., 2004*; *Kwon et al., 2018*; *Turner et al., 1997*; *Vilaseca et al., 2013*), with considerable heterogeneity ($I^2 = 76\%$, $p = 0.0002$) (Fig. 4G). Neither the funnel plot (Fig. S2E) nor Egger's test ($p = 0.423$) suggested publication bias.

## Sensitivity analysis
Sensitivity analysis was performed for various outcomes, including operating time, estimated blood loss (EBL), complications, urinary incontinence, CIC rate, and erectile function. This involved assessing the impact of individually excluding studies on the combined WMD or OR. The overall findings remained stable after the exclusion of any single study, except in the cases of operating time, EBL, and complications. Notably, removing *Kwon et al. (2018)* from the operating time analysis revealed significant intergroup differences ($p = 0.007$, $I^2 = 74\%$). Excluding *Huang et al. (2019)* and *Wang, Luo & Chen (2008)* led to the disappearance of heterogeneity in EBL ($I^2 = 0\%$, $p < 0.00001$) (*Huang et al., 2019*; *Wang, Luo & Chen, 2008*). Similarly, omitting *Bai et al. (2019)* clarified the heterogeneity in complications ($I^2 = 0$, $p = 0.04$). These findings are illustrated in Fig. S1 (A–C).

## Subgroup analysis
Subgroup analyses were conducted to identify sources of heterogeneity for several outcomes, including urinary continence, erectile function, and operating time, as detailed in Table 2. The heterogeneity in operative time was mainly attributed to urinary diversion (P4* =0.0004), while the source of heterogeneity for the other outcomes was linked to the type of surgery (P1* =0.002; P2*<0.0001; P3* =0.02).

## GRADE system
The GRADE system assessment showed that the quality of evidence was moderate for length of stay, recurrence rate, positive surgical margin rate, overall survival, CSS, and urinary incontinence at 6 and 12 months. The evidence quality was lower for operating time, complications, nighttime incontinence at 6 and 12 months, and erectile function after

Zhang et al. (2024), *PeerJ*, DOI 10.7717/peerj.18427

**Table 2  Subgroup analysis of continence, erectile function, and operating time.**

| | Daily Continence (6 month) | | | Nighttime Continence (6 month) | | | Erectile Function (>1 year) | | | Operating time | | |
|---|---|---|---|---|---|---|---|---|---|---|---|---|
| | No. of Trials | OR | P1 | No. of Trials | OR | P2 | No. of Trials | OR | P3 | No. of Trials | WMD | P4 |
| **Study Design** | | | 0.42 | | | 0.63 | | | 0.90 | | | 0.41 |
| prospective | 4 | 3.55 | 0.03 | 4 | 2.78 | 0.001 | 5 | 16.29 | <0.0001 | 1 | −6.00 | – |
| retrospective | 4 | 6.67 | 0.19 | 4 | 4.04 | 0.18 | 3 | 18.65 | 0.31 | 7 | −18.33 | <0.0001 |
| **Type Of Surgery** | | | 0.002[*] | | | <0.0001[*] | | | 0.02[*] | | | 0.80 |
| only nerve sparing | 5 | 2.55 | 0.39 | 5 | 1.69 | 0.40 | 6 | 7.97 | 0.07 | 4 | −10.97 | 0.0008 |
| other organ sparing | 3 | 10.51 | 0.46 | 3 | 10.51 | 0.46 | 2 | 148.12 | 0.19 | 4 | −18.42 | 0.0006 |
| **Publication Year** | | | 0.15 | | | 0.37 | | | 0.39 | | | 0.82 |
| ≤2013 | 4 | 2.75 | 0.21 | 4 | 2.31 | 0.03 | 5 | 9.87 | 0.03 | 2 | −19.92 | 0.07 |
| >2013 | 4 | 6.66 | 0.10 | 4 | 4.55 | 0.03 | 3 | 30.52 | 0.005 | 6 | −14.47 | <0.00001 |
| **Assessment Modality** | | | 0.91 | | | 0.44 | | | 0.07 | | | – |
| pads(0pad)/IIEF[a] | 5 | 4.16 | 0.01 | 5 | 3.49 | 0.0004 | 5 | 34.34 | 0.03 | – | – | – |
| pads(≤1pad)/other definitions[a] | 3 | 4.51 | 0.23 | 3 | 2.13 | 0.54 | 3 | 5.82 | 0.06 | – | – | – |
| **Urinary Diversion** | | | 0.68 | | | 0.73 | | | 0.63 | | | 0.0004[*] |
| only orthotopic neobladder | 4 | 5.14 | 0.01 | 4 | 3.92 | 0.0002 | 3 | 24.99 | 0.0002 | 3 | −11.50 | 0.13 |
| including other diversion[b] | 2 | 2.56 | 0.37 | 2 | 1.99 | 0.3 | 3 | 23.52 | 0.23 | 2 | 48.73 | 0.008 |
| not reported | 2 | 6.93 | 0.06 | 2 | 2.98 | 0.64 | 2 | 6.28 | 0.11 | 3 | −55.98 | 0.60 |
| **Summary** | 8 | 4.19 | 0.03 | 8 | 3.14 | 0.003 | 8 | 17.38 | 0.0002 | 8 | −16.99 | <0.00001 |

**Notes.**

[a] Variables described erectile function

[b] including ileal conduit or continent cutaneous

P[*] value for subgroup difference

IIEF, International Index of Erectile Function Questionnaire; WMD, weighted mean difference; OR, odds ratio.

1 year, and very low for EBL, CIC rate, and erectile function within 1 year, as presented in Table S4.

## DISCUSSION

In our systematic review and pooled analysis of 19 studies involving 2057 patients with muscle-invasive bladder cancer (MIBC) and high-risk non-muscle invasive bladder cancer (NMIBC), we explored the clinical safety and efficacy of organ-sparing cystectomy (OSC). While standard radical cystectomy (SRC) is effective in improving prognosis, it often compromises postoperative erectile function and urinary continence. Advances in laparoscopic and robotic technologies have made nerve and organ preservation more feasible, leading to increased adoption of OSC. However, the debate over OSC's safety and efficacy persists (*Patel et al., 2022*).

In our initial evaluation of surgical safety, there were no significant differences between the OSC and SRC groups in operation time, hospital stay, estimated blood loss, and complications. However, when excluding the study by *Kwon et al. (2018)* in the sensitivity analysis of operation time, a significant difference became evident between the groups ($p = 0.007$, $I^2 = 74\%$). This could be due to inconsistencies in surgical methods (robot-assisted OSC *versus* open SRC) (*Kwon et al., 2018*). Subgroup analysis indicated that heterogeneity mainly stemmed from variations in urinary diversion methods. However, as some studies did not detail their urinary diversion methods, these results should be interpreted cautiously. The sensitivity analysis also suggested potential instability in the outcomes for estimated blood loss and complications. Surgical safety is a complex metric, with some studies linking shorter OSC operation times to a reduced resection range (*Bai et al., 2019*; *Hernández et al., 2017*). It is noteworthy that OSC was developed after SRC, and surgeons generally have more experience with SRC. Factors such as the statistical methods of different hospitals, the skills of surgeons, and the type of surgery (robot-assisted or laparoscopic) significantly influence operation time, blood loss, and perioperative complications (*Bai et al., 2019*). The implementation of orthotopic neobladder in urinary diversion is also noted to potentially reduce surgery time and postoperative complications (*Yu et al., 2023*). In summary, these results on surgical safety are informative, yet they require further validation through large-scale, multicenter RCTs.

Secondly, the debate continues over whether preserving additional organs increases the risk of tumor recurrence and impacts long-term survival post-surgery (*Furrer et al., 2018*). A common perspective is that avoiding RC surgery may heighten the risk of local recurrence or metastasis, thus potentially reducing survival rates (*Hernández et al., 2017*). Our meta-analysis on tumor safety revealed no significant differences between the groups concerning recurrence rate, positive surgical margin rate, overall survival (5 years), and cancer-specific survival (5 years), presenting stable and reliable results ($I^2 = 0$). A primary concern with OSC is the potential risk of local recurrence and metastatic disease postoperatively. In their RCT on prostate-preserving cystectomy, *Abdelaziz et al. (2019)* found no difference in local recurrence rates between the SRC and OSC groups, with neither group showing distant metastasis after two years and no significant statistical difference. Studies with a minimum

of a 3-year follow-up have reported low local recurrence rates comparable to standard radical cystectomy (*Muto et al., 2004*; *Terrone et al., 2004*; *Vallancien et al., 2002*). These tumor outcomes align with conclusions from two prior review articles, suggesting that organ preservation does not compromise tumor outcomes (*Hernández et al., 2017*; *Veskimäe et al., 2017*). Factors such as preoperative age, clinical T stage, and neoadjuvant chemotherapy significantly influence postoperative tumor outcomes. Several of the included studies noted in their limitations that standard OSC may be more appropriate for patients with milder conditions and lower clinical T stages, potentially leading to selection bias (*El-Bahnasawy, Gomha & Shaaban, 2006*; *Huang et al., 2019*). Nevertheless, the majority of the studies reviewed did not report significant differences in age or preoperative clinical T stage between groups (Table S3), indicating that the oncologic safety outcomes in our study are robust and credible. Numerous studies demonstrate that bladder cancer responds effectively to platinum-based combined neoadjuvant chemotherapy, currently the gold standard treatment alongside radical cystectomy. Research involving neoadjuvant chemotherapy and radiotherapy showed no differences between groups, underscoring the tumor safety we examined as having substantial reference value.

Thirdly, regarding clinical efficacy, the prevailing view is that OSC enhances functional outcomes, primarily through the preservation of neurovascular bundles (NVB) that control sexual function and micturition by saving nerves or various pelvic organs. In males, erectile function depends on the parasympathetic innervation of the cavernous nerves, which traverse the pelvis and prostatic plexus to the penis. These nerves are anatomically close to the bladder, seminal vesicles, prostate, and urethral sphincter (*Dean & Lue, 2005*). Similarly, in women, pelvic nerves also play a crucial role in vaginal sensation and lubrication. In addition, the pelvic parasympathetic nerves, lumbar sympathetic nerves, and pudendal nerves, which regulate micturition, are in proximity to these structures. Radical cystectomy (RC) entails the complete removal of the bladder and surrounding structures, posing considerable risks to these nerves (*Yoshimura & Chancellor, 2003*). In our meta-analysis, the OSC group significantly outperformed the SRC group in both daytime and nighttime urinary continence, in the short-term (6 months) and long-term (1 year) (*Abdelaziz et al., 2019*; *Kessler et al., 2004*; *Turner et al., 1997*; *Wang, Luo & Chen, 2008*). Similarly, in the long-term results data from studies with follow-up periods exceeding 5 years, OSC continued to show a significant advantage in urinary continence. *Furrer et al.'s (2018)* research suggests that OSC combined with orthotopic neobladder offers improved long-term urinary control, particularly in older patients. CIC rates showed no significant differences between the groups, with stable results in the sensitivity analysis, indicating OSC's advantage in urinary control, though some OSC patients still require regular clean intermittent catheterization. For erectile function, the OSC group demonstrated significant benefits both within the first year and after one year post-surgery (within 1 year: 27.52 [2.58, 294.07]; after 1 year: 17.38 [5.42, 55.70]). Sensitivity analyses for daytime and nighttime urinary control at 6 months and erectile function after one year yielded stable results, with subgroup analysis clarifying heterogeneity due to different surgical techniques. Notably, in SRC group studies post-surgery, several patients exhibited normal erectile and urinary control functions, suggesting that the postoperative regulation of these functions is not

exclusively related to nerve or organ preservation. Postoperative erectile dysfunction and urinary incontinence involve a complex array of pathophysiological factors, and currently, data are insufficient for a comprehensive study of these.

This study performed a systematic evidence-based analysis of OSC, but it is important to acknowledge certain limitations in the current research. First, OSC is not commonly practiced clinically and is usually reserved for patients with a strong preference for preserving sexual and urinary control functions, which could lead to selection bias. Second, our pooled analysis incorporated only one prospective randomized study, predominantly featuring retrospective or prospective cohort studies, which may not adequately control for confounding factors. Furthermore, significant heterogeneity was noted in some outcomes. Although sensitivity and subgroup analyses were conducted to evaluate result stability, the analyses for EBL and complications remained unstable, and the exact sources of heterogeneity are not fully understood. Lastly, due to limitations in the available raw data, further stratification by pathological stage might reveal differences in oncological outcomes between ORC and SRC. Similarly, the included studies also lacked assessments of female sexual function.Therefore, the results of this meta-analysis should be cautiously interpreted due to these potential confounding factors.

Despite these limitations, the strength of our research lies in providing a systematic and comprehensive analysis of the clinical safety and efficacy of organ-sparing cystectomy. The stability of the sensitivity analyses for most outcome measures, along with the GRADE system evaluation, suggests that these findings are valuable references for clinical treatment. Urologists may more often consider OSC based on their experience and specific patient factors. The demonstrated benefits in quality of life may influence clinical decision-making, encouraging a tailored approach to patient care. Future research should include more well-designed, large-scale prospective randomized studies with long-term follow-up to better compare the clinical safety and efficacy of OSC and SRC.

## CONCLUSION

Comprehensive analysis indicates that compared to SRC, OSC can significantly improve postoperative erectile function and urinary continence without significant differences in surgical and oncological safety between the two groups. Despite limited clinical practice and potential selection bias, urologists may still consider OSC more based on their experience and specific patient factors. The demonstrated benefits in quality of life may influence clinical decision-making, encouraging a tailored approach to patient care.

### Funding

This work was supported by the Shenzhen Medical research special fund project; project approval number: A2302048. The funders had no role in study design, data collection and analysis, decision to publish, or preparation of the manuscript.

### Grant Disclosures

The following grant information was disclosed by the authors:

The Shenzhen Medical research special fund project: A2302048.

## Competing Interests

The authors declare there are no competing interests.

## Author Contributions

- Yi Zhang conceived and designed the experiments, performed the experiments, authored or reviewed drafts of the article, and approved the final draft.
- Lei Peng conceived and designed the experiments, performed the experiments, authored or reviewed drafts of the article, and approved the final draft.
- Yang Zhang performed the experiments, analyzed the data, authored or reviewed drafts of the article, and approved the final draft.
- Hangxu Li analyzed the data, authored or reviewed drafts of the article, and approved the final draft.
- Songbei Li analyzed the data, prepared figures and/or tables, and approved the final draft.
- Shaohua Zhang analyzed the data, authored or reviewed drafts of the article, and approved the final draft.
- Jianguo Shi conceived and designed the experiments, authored or reviewed drafts of the article, and approved the final draft.

## Data Availability

This is a systematic review/meta-analysis.

## Supplemental Information

Supplemental information for this article can be found online at http://dx.doi.org/10.7717/peerj.18427#supplemental-information.

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
