# Peer review of "Clinical efficacy and safety of organ-sparing cystectomy: a systematic review and meta-analysis"

_PeerJ, doi:10.7717/peerj.18427_

## Round 0.1 · original submission · Major Revisions

The reviewers feel the article is well-structured and the discussion is comprehensive. However, based on the the concerns noted by the reviewers, I have decided that this review needs further modifications before being accepted. Kindly address all the suggestions/concerns by the reviewers.

Reviewer 1 ·

Basic reporting

Overall, this is a comprehensive review of organ-sparing cystectomy (ORC) in male and female patients. ORC is an active area of innovation and the topic is relevant and noteworthy.

I would rename “clinical efficacy” (section 3.4) to “outcome measures” or something similar.

The PRISMA diagram and forest plot figures are very well-constructed.

Experimental design

Would be good practice to include the date of first study in methods (rather than just saying “up to February 2024.”

There is independent author review of PedMed sources, a critical component of the manuscript.

Validity of the findings

Quality assessment statement included in the methods.

The focus on group heterogeneity as a statistical measure is unique and adds to the overall findings.

Additional comments

Very interesting findings overall, with no differences in LOS, EBL, operating time, and complication rate. It would be good to specify in result the exact “complication rate” (i.e. 90-day complication rate vs. overall complication rate; major vs. minor).

No difference in recurrence rate, positive surgical margins, OS, or CSS. These are key findings.

Significant improvement in urinary continence and nighttime continence for ORC. Erectile function was also improved. Recommend seeing if any of these studies had long-term outcome data (i.e. past 6 months, for incontinence, or 1 year, for erectile function). Line 162 comments on publication bias – this is better-suited for discussion.

The discussion is well-constructed and clearly stated. Citation of specific studies adds to the value of the manuscript.

One limitation of the study which could potentially be investigated depending on the raw data is that it did not stratify by pathologic stage. Differences may be detected oncologic outcomes between ORC and SRC if stratified by stage. This could be an additional acknowledgment in the discussion.

Overall, this is a strong paper that is suitable for publication with minimal revisions.

·

Basic reporting

### Comments and Suggestions for Improvement

#### Title:
**Current Title**: "Clinical Efficacy and Safety of Organ-Sparing Radical Cystectomy"
**Suggestions**:
- Consider rephrasing for clarity: "Clinical Efficacy and Safety of Organ-Sparing Cystectomy"
- Ensure the title reflects the scope and key findings of the study.

#### Abstract:
1. **Background**:
- The background is concise and clear. However, it could benefit from a brief mention of why OSC might be preferable in certain patient populations.
- **Suggestion**: Add a line on the potential quality of life improvements OSC aims to address.

2. **Methods**:
- Clear and well-structured. The adherence to PRISMA and AMSTAR guidelines is well noted.
- **Suggestion**: Include a brief description of the types of studies included (RCTs, cohort studies, etc.)

3. **Results**:
- Comprehensive and covers all the key findings.
- **Suggestion**: Highlight the statistical significance of the benefits observed in erectile function and urinary continence.

4. **Conclusions**:
- Summarizes the key findings well but could emphasize the potential clinical implications more.
- **Suggestion**: Add a statement on how these findings might influence clinical decision-making.

#### Introduction:
1. **Current Introduction**:
- Provides a good overview of bladder cancer and the traditional treatments.
- **Suggestions**:
- Include more recent statistics or references if available.
- Clearly state the hypothesis or primary question the study aims to address.
- Briefly mention why previous studies have been inconclusive or what gaps in knowledge this study addresses.

#### Materials and Methods:
1. **Protocol**:
- Well-documented adherence to guidelines.
- **Suggestion**: Ensure that the registration number and any additional protocol details are easy to locate.

2. **Literature Search**:
- Comprehensive search strategy.
- **Suggestion**: Detail the search terms used in the main text rather than just in a supplementary table for better transparency.

3. **Identification of Eligible Studies**:
- Clear inclusion and exclusion criteria.
- **Suggestion**: Clarify the rationale behind some of the exclusion criteria (e.g., why studies on prostate and uterus cancer were excluded).

4. **Data Extraction**:
- Clear and thorough.
- **Suggestion**: Include any software or tools used for data extraction and analysis.

5. **Quality Assessment**:
- Detailed and systematic.
- **Suggestion**: Provide a brief summary of the findings from the quality assessment in the main text.

6. **Statistical Analysis**:
- Comprehensive description of methods.
- **Suggestion**: Justify the choice of statistical models and any assumptions made.

#### Results:
1. **Literature Search and Study Characteristics**:
- Clear presentation of the study selection process.
- **Suggestion**: Consider adding a summary table in the main text to highlight key characteristics of the included studies.

2. **Surgical Safety**:
- Well-detailed findings.
- **Suggestion**: Discuss any potential biases that could have influenced these results.

3. **Oncological Safety**:
- Clear and thorough.
- **Suggestion**: Emphasize the clinical significance of these findings and compare them to previous literature.

4. **Clinical Efficacy**:
- Comprehensive analysis.
- **Suggestion**: Discuss the potential impact of the findings on patient quality of life in more detail.

#### Discussion:
1. **Current Discussion**:
- Provides a balanced discussion of the findings.
- **Suggestions**:
- Address any limitations of the included studies more explicitly.
- Discuss the implications of these findings for future research and clinical practice.
- Consider including a section on the strengths of the study.

2. **Subgroup Analysis and Sensitivity Analysis**:
- Well-detailed but could be more concise.
- **Suggestion**: Summarize key findings from these analyses in a more streamlined manner.

3. **GRADE System**:
- Important for assessing evidence quality.
- **Suggestion**: Provide a brief explanation of what the GRADE system is and why it's used.

#### Conclusion:
1. **Current Conclusion**:
- Summarizes the main findings well.
- **Suggestions**:
- Reinforce the clinical relevance of OSC and its potential benefits.
- Mention any plans for future research or follow-up studies.

#### References:
1. **Current References**:
- Comprehensive and up-to-date.
- **Suggestions**:
- Ensure consistency in citation format.
- Verify that all references are accessible and relevant.

#### Overall:
1. **Presentation**:
- The manuscript is well-organized and thorough.
- **Suggestions**:
- Consider a final proofread for grammar, spelling, and typographical errors.
- Ensure all tables and figures are clear and well-labeled.
- Include any additional visual aids or summary tables in the main text for better clarity.

This manuscript presents valuable findings on the clinical efficacy and safety of organ-sparing cystectomy. By addressing these suggestions, the manuscript can be further strengthened to provide clear and impactful insights for publication.

Experimental design

As above

Validity of the findings

As above

Additional comments

As above

·

Basic reporting

no comment

Experimental design

no comment

Validity of the findings

138-154 - Oncologic Safety section: The results and comparisons here are done well. It could benefit from the inclusion of more information regarding staging of the tumors if there is any variance whatsoever between the patient populations that receive one surgery versus the other.


235-241: This section could include more information as to how the various studies analyzed accounted for the potential variance in preoperative tumor staging between the two procedures being studied. Especially considering the comments by the authors that typically OSC receiving patients typically have milder disease burdens.

---

## Round 0.2 · Minor Revisions

Dear Dr. Shi,

Congratulations, our reviewers have checked your submission and are happy with the responses. However, before we can accept your manuscript for publication, I would recommend you to address the few minor suggestions raised by Reviewer 3.
Thank you!

Reviewer 1 ·

Basic reporting

All issues from the initial submission were addressed and it is appropriate for publication.

Experimental design

All issues from the initial submission were addressed and it is appropriate for publication.

Validity of the findings

All issues from the initial submission were addressed and it is appropriate for publication.

Additional comments

All issues from the initial submission were addressed and it is appropriate for publication.

·

Basic reporting

.

Experimental design

.

Validity of the findings

.

Additional comments

The manuscript titled "Clinical Efficacy and Safety of Organ-Sparing Cystectomy" offers a comprehensive analysis of the benefits and risks associated with organ-sparing approaches to cystectomy compared to standard radical cystectomy (SRC). Here are some key points and areas to consider before publication:

1. **Strengths**:
- **Detailed Methodology**: The systematic review adheres to the PRISMA 2020 guidelines, ensuring transparency and reproducibility in the study selection and data extraction process.
- **Clear Focus on Key Outcomes**: The manuscript appropriately compares critical metrics such as surgical safety, oncological outcomes, and functional efficacy (urinary continence, erectile function). These are essential for evaluating the clinical relevance of organ-sparing cystectomy (OSC).
- **Comprehensive Data Analysis**: The statistical evaluation of outcomes is robust, using odds ratios and weighted mean differences with appropriate confidence intervals. Sensitivity analyses and heterogeneity assessments further strengthen the conclusions.

2. **Potential Improvements**:
- **Clarify Patient Selection**: Although the manuscript mentions potential selection bias, it would benefit from further clarification on how patients were selected for OSC versus SRC, particularly given that OSC may be more appropriate for specific patient populations (e.g., younger, less advanced disease).
- **Long-Term Follow-Up**: The results focus significantly on the short-term benefits (6-12 months) regarding functional outcomes like erectile function and continence. Further emphasis on long-term oncological safety (beyond 5 years) would solidify conclusions, especially considering concerns over tumor recurrence.
- **Address Heterogeneity**: Although subgroup and sensitivity analyses were conducted, significant heterogeneity in certain outcomes (e.g., operating time, blood loss, complications) suggests that variations in surgical technique (robotic vs. open surgery) may have influenced the results. Providing additional granularity on the type of surgery performed could improve the manuscript's reliability.

3. **General Comments**:
- **Conservative Interpretation**: While the study finds no significant differences in recurrence or survival rates between OSC and SRC, it’s important to frame the results conservatively, considering the limited number of randomized controlled trials and the potential for selection bias.
- **Functional Outcomes**: The findings on functional outcomes, particularly erectile function and continence, strongly favor OSC, which is a key selling point. This supports OSC as a viable option for quality-of-life preservation, especially for younger patients or those desiring sexual function preservation.

In conclusion, this manuscript contributes valuable data to the ongoing debate regarding organ-sparing cystectomy. Addressing the limitations in patient selection and the heterogeneity of results could further strengthen its impact. The overall findings suggest that OSC offers meaningful benefits in functional outcomes without compromising oncological safety, making it a promising alternative to SRC in select patient populations.

·

Basic reporting

Overall well written and researched.

A few notes:
- I'm not sure what "in situ transfer techniques" refers to in line 67
- In lines 111-116 would avoid capitalizing words that are not proper nouns
- Lines 274-278 -- the role of chemotherapy and radiation in this paper are unclear
- The figure label for figure 4 does not include full descriptions of F and G
- If the events in Figure 4 are continence, the labels under the Forest plot (favours OSC vs favours SRC) appear to be switched. For example, the paper states that continence was improved in OSC however Figure 4A states that continence favors SRC.
- Would list "NOS" in the abbreviations under Table 1

Experimental design

This is an important question with limited data available, as the authors correctly highlight. Methods are clear and concise.

Validity of the findings

Conclusions are appropriately stated. The limitations are well written and add to the credibility of the paper, especially the point regarding selection bias.

Additional comments

This paper organ sparing cystectomy in both male and female patients.

Recommend adding a sentence around line 285 discussing the role of pelvic nerves in vaginal sensation and lubrication.

I would be curious to see the breakdown of male/female patients in OSC and SRC arms.

Did any of the studies examined have outcome measure of female sexual function? If not, I would mention this as a limitation.

---

## Round 0.3 · accepted · Accept

Dr. Shi, congratulations. Your article is now accepted for publication.